# Beaver dams overshadow climate extremes in controlling riparian hydrology and water quality

Christian Dewey[1], Patricia M. Fox [2], Nicholas J. Bouskill [2], Dipankar Dwivedi[2], Peter Nico[2] & Scott Fendorf [1] ✉

Hydrologic extremes dominate chemical exports from riparian zones and dictate water quality in major river systems. Yet, changes in land use and ecosystem services alongside growing climate variability are altering hydrologic extremes and their coupled impacts on riverine water quality. In the western U.S., warming temperatures and intensified aridification are increasingly paired with the expanding range of the American beaver—and their dams, which transform hydrologic and biogeochemical cycles in riparian systems. Here, we show that beaver dams overshadow climatic hydrologic extremes in their effects on water residence time and oxygen and nitrogen fluxes in the riparian subsurface. In a mountainous watershed in Colorado, U.S.A., we find that the increase in riparian hydraulic gradients imposed by a beaver dam is 10.7–13.3 times greater than seasonal hydrologic extremes. The massive hydraulic gradient increases hyporheic nitrate removal by 44.2% relative to seasonal extremes alone. A drier, hotter climate in the western U.S. will further expand the range of beavers and magnify their impacts on watershed hydrology and biogeochemistry, illustrating that ecosystem feedbacks to climate change will alter water quality in river systems.

The exchange of water and solutes between river channels and the shallow subsurface (hyporheic exchange) exerts a predominant control on biogeochemical exports from mountain floodplains and is a primary determinant of riverine water quality. Hydrologic extremes, such as rapid snowmelt and severe rain events, alter water residence time and solute exchange rates across mountain hyporheic zones[1–3]. As climate change reshapes temperature and precipitation regimes throughout mountain watersheds, shifts in the duration, magnitude, and timing of hydrologic extremes will drive coupled shifts in riparian fluxes of nutrients and contaminants, altering riverine water quality[4–7].

In the western U.S., warming temperatures and intensified aridification are occurring alongside the resurgence of the American beaver and, consequently, a proliferation of beaver dams. Ecosystem management practices have largely returned beaver populations to their historical range[8,9], and in so doing have restored the ecosystem services that beavers provide, including increased water storage and residence times at the catchment scale[10]; increased hyporheic exchange of contaminants and nutrients[11,12]; and reduced peak discharge[13,14]. As temperatures warm and precipitation decreases throughout the western U.S., the range and density of beaver populations is expected to expand[15,16]. Thus, not only will climatic factors directly impact riparian hydrologic and biogeochemical cycles by shifting the magnitude and timing of riparian hydrologic extremes; they will also compound the impacts of beaver dams on these cycles, driving potent a climate feedback in ecosystems services. However, the magnitude of the feedback is unknown. Continued expansion of beaver populations may drive hydrologic and biogeochemical changes in mountain river systems that rival the changes imposed by shifts in temperature and precipitation alone.

[1]Earth System Science Department, Stanford University, Stanford, CA 94305, USA. [2]Earth and Environmental Sciences Area, Lawrence Berkeley National Laboratory, Berkeley, CA 94720, USA. ✉e-mail: fendorf@stanford.edu

In this study, we compare the effects of beaver dams and climatic hydrologic extremes on riparian fluxes of oxygen and nitrogen and their impacts on riverine water quality. Excess nitrate or ammonium, the predominant reactive nitrogen species in freshwater, is a persistent global threat to water quality[17,18]. River headwaters are particularly effective at regulating downstream loading of reactive nitrogen, with hyporheic exchange playing a central role in determining exports of reactive nitrogen from mountain watersheds[19]. The hyporheic zone functions as a source or sink of reactive nitrogen depending on water residence time and nitrogen transformation rates within the shallow subsurface[20–22]. Using field measurements and reactive transport modeling, we compared the hydrologic and biogeochemical impacts of beaver dams and climatic hydrologic extremes on hyporheic reactive nitrogen cycling in the headwaters of the Colorado River. We find that the hydraulic gradients imposed by beaver dams greatly exceed the gradients imposed by climate extremes, leading to shortened water residence times and increased oxygen and nitrogen fluxes across hyporheic zones. We reveal that beaver dams overshadow climatic hydrologic extremes in controlling the exports of reactive nitrogen from mountain riparian zones and, further, that management practices and ecosystem feedbacks to climate change can generate ecosystem services that overcome the detrimental effects of climate change.

## Results and discussion
### Seasonal and beaver-driven hydrologic extremes
Our study focuses on a meandering reach of the East River, a main tributary to the Colorado River, near Crested Butte, Colorado, USA. In 2018, historic low-water conditions occurred across the western U.S., foreshadowing the low-water extremes expected with continued warming and intensified aridification in the region[23–25]. In contrast, 2019 was a moderately high-water year. Hydrologic conditions throughout the East River watershed reflected the regional trends in 2018 and 2019: between 1935 and 2021, peak discharge fell below the

2018 level only three times, reflecting the historic low-water conditions in that year, and exceeded the 2019 level 14 times (Supplementary Fig. 1). Over these two contrasting water years, we compared water levels and associated biogeochemical cycles in a riparian area bounded by the East River. We installed an array of pressure transducers throughout the riparian area, including in the river channel, to measure hourly water levels at the site across hydrologic transitions (Supplementary Fig. 2). We also installed a transect of piezometers aligned with the general direction of subsurface flow, from which we collected water samples one to three times a week between May and October in 2018 and 2019 (Supplementary Fig. 2).

In the summer of 2018, amid historic low-water conditions, a beaver dam was built in our study reach across the main channel of the East River (Supplementary Fig. 2), which allowed us to assess the effects of the dam on hydrologic and biogeochemical processes within the adjacent hyporheic zone. Construction of the dam began between July 26 and July 30, 2018, and continued until October 5, 2018, when the dam was destroyed. As construction of the dam proceeded, upstream water levels steeply increased, while downstream water levels did not, resulting in a large increase in the hydraulic gradient, $i$, across the riparian hyporheic zone (Fig. 1). Before the dam was destroyed, it imposed a maximum gradient of 0.017 m/m across adjacent hyporheic sediments and soils, an increase of 161.5% relative to the average gradient prior to construction of the dam. The maximum gradient imposed by the beaver dam dwarfed the maximum gradients imposed by snowmelt-driven hydrologic extremes in both 2018 (0.0073 m/m; 12.3% increase relative to pre-dam average) and 2019 (0.0061 m/m; 15.1% increase relative to the yearly average) (Fig. 1).

### Redox zonation during hydrologic extremes
The beaver dam more than doubled the extent of the riparian aerobic zone relative to snowmelt-driven extremes. Pairing hydrologic observations with measurements of porewater pH, dissolved oxygen (DO),

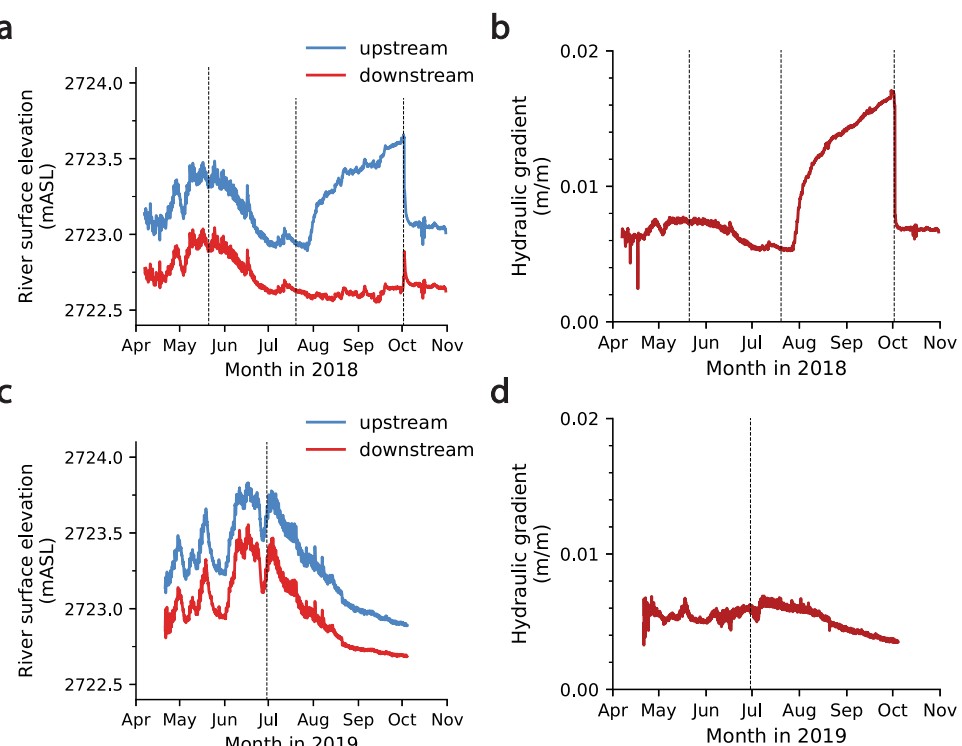

**Fig. 1 | River surface elevations and hydraulic gradients. a** Upstream (blue) and downstream (red) river surface elevations at the study site from April 7 through October 31, 2018. **b** The hydraulic gradient across the floodplain from April 7 through October 31, 2018. **c** River surface elevations from April 21 through October 2, 2019. **d** The hydraulic gradient across the floodplain from April 21 through October 2, 2019. Dashed black lines indicate dates for which profiles of dissolved oxygen (DO), nitrate ($NO_3^-$), and ammonium ($NH_4^+$) are shown in Fig. 2.

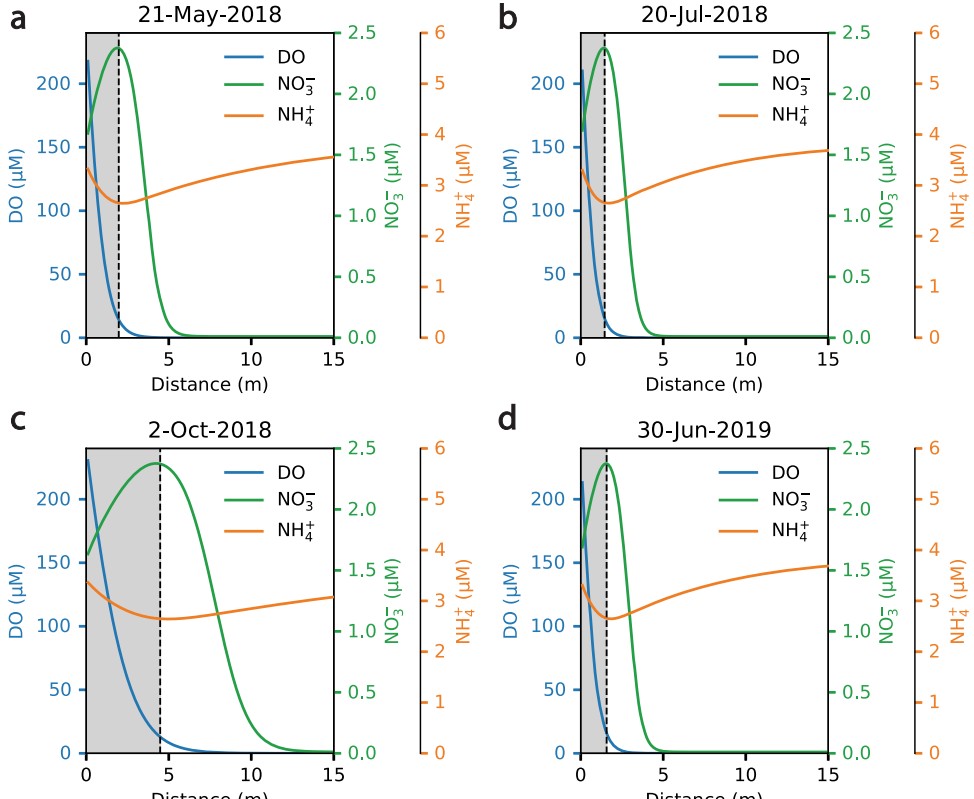

**Fig. 2 | Flow path concentrations of dissolved oxygen and reactive nitrogen.** Modeled concentration profiles of dissolved oxygen (DO, blue line), nitrate ($NO_3^-$, green line), and ammonium ($NH_4^+$, orange line) along a riparian flow path at: **a** peak river discharge in 2018, a historically low water year; **b** baseflow conditions in 2018; **c** the maximum beaver dam water levels in 2018; and **d** peak discharge in 2019, a high-water year. Within each panel, gray shading indicates the extent of flow path over which DO concentrations are primarily determined by advection (i.e., where the Damköhler number for DO, $Da_{DO}$, is less than 1). The dashed lines correspond to the point at which DO consumption overtakes advection as the primary determinant of DO concentration (i.e., where $Da_{DO} = 1$). The unshaded regions denote where DO consumption determines DO concentrations (i.e., where $Da_{DO} > 1$). Figure 1 indicates the water levels and hydraulic gradients corresponding to these concentration profiles. Model validation results are shown in Supplementary Figs. 13–15.

nitrate, ammonium, calcium, and dissolved carbon, we developed a reactive transport model to assess transient redox zonation across the monitored transect at our site. The biogeochemical reaction network contains rate formulations for aerobic microbial respiration, ammonification (mineralization), nitrification, denitrification, calcite precipitation and dissolution, and acetogenesis. Reaction parameters were constrained to values used in previously published studies and, where necessary, tuned to yield output consistent with our observations (sources and values of parameters are shown in Supplementary Tables 1–3). Details of the model formulation appear in the Methods section. We then employed a Damköhler analysis for DO ($Da_{DO}$) to delineate the aerobic and anaerobic zones along the flow path, defining the location on the flow path where $Da_{DO} = 1$ (i.e., where transport and reaction processes equally influence DO concentrations) as the transition from aerobic to anaerobic conditions. Beyond this location, microbial demand for DO outpaces its supply (via advection) and DO is rapidly depleted. At the maximum hydraulic gradient imposed by the beaver dam, the aerobic zone extends 4.37 m into the hyporheic zone (Fig. 2). In comparison, during the snowmelt-driven extremes, the aerobic zone extends only 1.92 m and 1.57 m in 2018 and 2019, respectively (Fig. 2). Extreme seasonal gradients only marginally increase advection along the flow path, whereas the beaver-driven gradient increases advection substantially, leading to increased supply of DO and a pronounced expansion of the riparian aerobic zone.

The expansion of the aerobic zone is paired with a narrowing of the denitrification zone. Within the expanded aerobic zone, nitrate concentrations increase due to mineralization of N-bearing soil organic matter (N-SOM) and nitrification[20,21]. Simultaneously, the

presence of oxygen suppresses the use of nitrate as an electron acceptor, and the denitrification front is extended farther along the flow path[26]. This is reflected in the DO, nitrate, and ammonium profiles along the representative flow path (Fig. 2 and Supplementary Figs. 3 and 4). Where DO concentrations are predominantly influenced by advection (Da < 1), nitrate concentrations increase, and ammonium concentrations decrease. Nitrate concentrations peak at the point along the flow path where DO consumption overtakes advection as the primary determinant of DO concentrations (i.e., where $Da_{DO} = 1$) (Fig. 2 and Supplementary Figs. 3 and 4). Immediately beyond this point, unutilized DO is rapidly depleted and denitrification becomes viable. Denitrification then predominates along the flow path until nitrate is consumed, and ammonium concentrations rebound due to ammonification of N-bearing soil organic matter[27].

The relative shifts in redox zonation between seasonal and beaver-driven hydrologic extremes is insensitive to parameterization of microbial mechanisms and remains proportional to shifts in the hydraulic gradient. We performed a set of Monte Carlo simulations (5000 realizations) in which we varied the rates of aerobic respiration and nitrification without changing other components of the model. Distributions for respiration and nitrification rates were derived from previously published studies[2,28–31], and realizations of the model were formulated by randomly sampling the distributions. For each realization, we determined the distance to the anaerobic zone (i.e., position of $Da_{DO} = 1$) at seasonal high-water and historic low-water conditions in 2018, as well at the beaver-induced maximum gradient. Distributions of the distances to $Da_{DO} = 1$ for these conditions are shown in Supplementary Fig. 5. While the overall rate of oxygen consumption

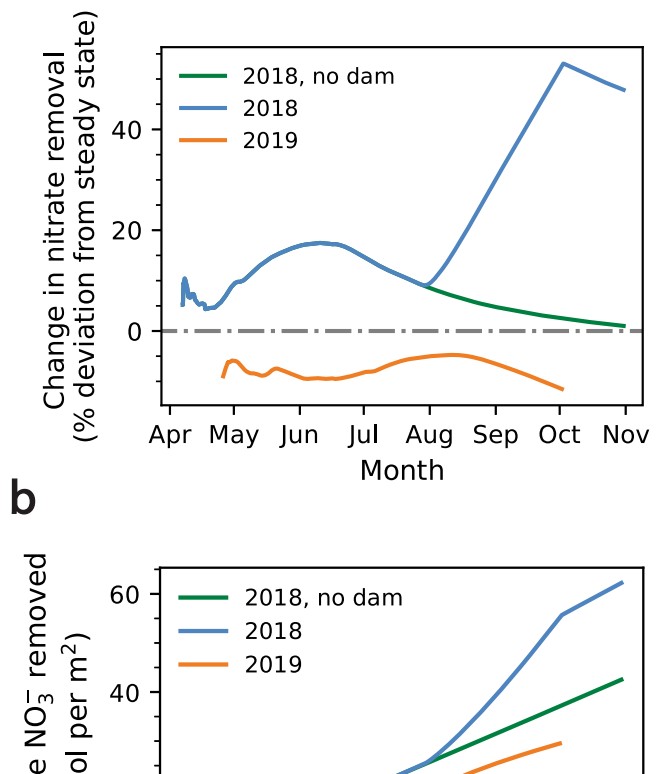

**Fig. 3 | Comparison of nitrate removal across hydrologic conditions. a** Change in nitrate removal, measured as percent deviation from steady-state hydrologic conditions, along a representative riparian flow path (58 m) under 2018 hydrologic conditions, with and without the beaver dam (blue and green lines, respectively), and 2019 hydrologic conditions (orange line). The mean of the initial hydraulic gradients in 2018 and 2019 was used to simulate steady-state conditions. **b** Cumulative nitrate removed in 2018 and 2019, expressed as mol $NO_3^-$ removed per m² cross sectional discharge area.

affects the magnitude of the distance to $Da_{DO} = 1$, the relative impact of the beaver dam on this distance is unchanged for a given parameter set and is proportional to the relative shift in hydraulic gradient between the extremes.

## Effects on transient fluxes and cumulative nitrate removal

As the transient hydrologic extremes shift advection and redox processes along the 58 m flow path at our study site, hyporheic fluxes of dissolved oxygen and nitrate are also altered. To determine the impacts of seasonal and beaver-driven hydrologic conditions on nitrate fluxes, we tracked hyporheic nitrate fluxes in 2018 and 2019 and compared them to nitrate fluxes under a steady-state hydrologic condition. In the steady-state simulation, the hydraulic gradient was set to the mean of the hydraulic gradients on the first days in 2018 and 2019 when the river channel was free of ice (April 7 and April 21, respectively). We also simulated and determined nitrate fluxes using water levels with the beaver dam hypothetically removed. We then quantified nitrate removal, defined as the difference in nitrate flux at

the upstream and downstream boundaries, along the flow path over the course of each hydrologic period.

Transient increases in advection also increase nitrate fluxes into the hyporheic zone. Because the flow path retains a region in which denitrification is viable—even at the maximum gradient imposed by the beaver dam—inflowing nitrate is ultimately removed along the flow path, and an increase in nitrate influx is coupled with an eventual increase in nitrate removal. Nitrate removal generally follows the hydraulic gradient, but lags by 4–5 weeks (Figs. 1 and 3), reflecting that inflowing water must first travel across the aerobic zone, where denitrification is inhibited and, further, where mineralization and nitrification increase porewater nitrate concentrations. Thereafter, the flow reaches the denitrification front, at which point nitrate is reduced and removed from porewater. In 2018, as water levels decline from the snowmelt-driven peak in late May to baseflow conditions in mid-June, nitrate removal along the flow path increases 17.5% relative to steady state, reflecting the modest increase in gradient at seasonal high water and the associated increase in the flux of nitrate into the denitrification zone (Fig. 3). Following construction of the beaver dam and the associated increase in nitrate advection, nitrate removal increases to 53.1% of steady-state levels (Fig. 3). After the dam is destroyed, nitrate removal decreases slowly despite the rapid return to pre-dam hydrologic conditions, due to nitrate produced within the aerobic zone before the dam was destroyed (Fig. 3). In 2019, maximum nitrate removal, which occurs in August following the maximum hydraulic gradient in July, is 4.8% less than nitrate removal under the steady state condition, and 22.3% and 57.9% less than nitrate removal at, respectively, the snowmelt- and beaver-driven maxima of 2018 (Fig. 3).

Overall, the transient effects of the beaver dam increase cumulative nitrate removal by 44.2% relative to conditions without the dam. From April 7 and October 31, 2018, 0.062 mol $NO_3^-$ are removed per m² of cross-sectional discharge area, whereas in the absence of the dam, only 0.043 mol $NO_3^-$ are removed per m² (Fig. 3). Normalized to the duration of the observation period (207 d), this equates to an average of $3.0 \times 10^{-4}$ and $2.1 \times 10^{-4}$ mol $NO_3^-$ removed per day per m² of cross-sectional area, with and without the dam, respectively. In contrast, between April 21 and October 6, 2019, a period of 168 days, 0.030 mol $NO_3^-$ are removed per m² of cross-sectional area, an average of $1.8 \times 10^{-4}$ mol $NO_3^-$ per day per m² discharge area (Fig. 3).

## Nitrate removal across the floodplain

As warming temperatures and intensifying aridification increase the range and density of beaver populations, the impacts of beaver dams on hyporheic nitrate fluxes will alter nitrate exports at the watershed scale. Already, beaver dams are common throughout the East River watershed. Using Google Earth imagery from October 2019, we identified 18 beaver dams within the 86 km² area of the watershed (Supplementary Figs. 6 and 7). This is likely an underestimation of the true number of dams constructed in 2019, as we only counted dams that were unambiguous in the satellite imagery. Given the prevalence of beaver dams in this watershed, their hydrologic and biogeochemical impacts are likely to affect nitrate fluxes regionally.

Yet, the impact of beaver dams on hyporheic nitrate fluxes depends on the lengths of affected flow paths, as path length is a primary determinant of hydraulic gradient. Thus, to assess the impacts of beaver dams on redox zonation and nitrate fluxes across the watershed, it is first necessary to determine the distribution of hyporheic flow path lengths. We determined an approximate distribution of flow path lengths within meandering regions of the East River floodplain using Google Earth™ satellite imagery (Supplementary Figs. 8 and 9; process of flow path selection described in detail in the Methods section). These regions are representative of average valley grade and river sinuosity across the meandering regions of the East River watershed, an assessment based on a digital elevation model

**a**

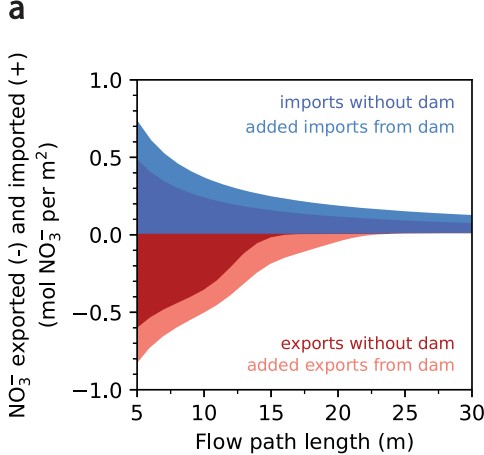
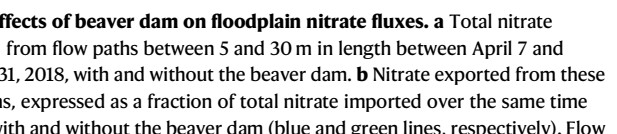

**b**

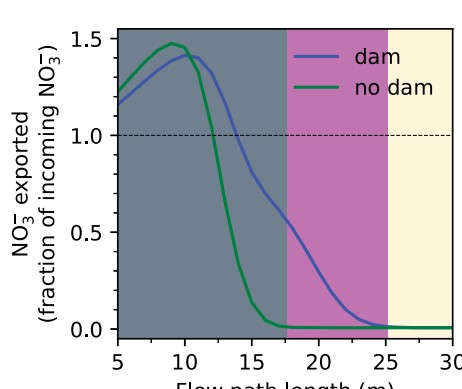

**Fig. 4 | Effects of beaver dam on floodplain nitrate fluxes. a** Total nitrate exported from flow paths between 5 and 30 m in length between April 7 and October 31, 2018, with and without the beaver dam. **b** Nitrate exported from these flow paths, expressed as a fraction of total nitrate imported over the same time period, with and without the beaver dam (blue and green lines, respectively). Flow paths shaded in gray are those that export nitrate when the dam is not present. Flow paths shaded in magenta are those that export nitrate only in the presence of the beaver dam. Flow paths shaded in beige do not export nitrate under either condition.

of the East River watershed[32]. To evaluate the impact of path length on hyporheic nitrate fluxes and redox zonation, we created reactive transport models for each flow paths between the minimum and maximum of the distribution (5 to 70 m) at 0.5 m increments. In each individual model, the reaction networks and boundary conditions (both hydrologic and geochemical) were identical to those used in the model of our site. We then simulated oxygen and reactive nitrogen transport across the range of flow paths and examined redox zonation at the seasonal and beaver-driven hydrologic extremes. During seasonal extremes, only flow paths shorter than 10.2 m, or approximately 3.6% of flow paths within the East River floodplain, are entirely aerobic and net nitrate exporters (Supplementary Fig. 8). In contrast, at the maximum beaver-driven gradient, flow paths up to 16.1 m in length (14.2% of flow paths) are entirely aerobic and export additional nitrate to the river (Supplementary Fig. 8). For flow paths longer than 16.1 m (85.8% of flow paths), the aerobic zone expands in response to the beaver dam; however, these flow paths remain net-denitrifying (although with a diminishing reach) during both the seasonal and beaver-driven hydrologic extremes (Supplementary Fig. 8).

Assuming beavers equally construct dams near long and short flow paths, most beaver dams predominantly affect flow paths that are net-denitrifying, even at the maximum beaver-driven gradient. However, beaver dams are often transient features within a watershed, and thus their impact on watershed-scale nitrate removal over the course of a hydrologic year is uncertain. To assess the impact of beaver dams on cumulative nitrate removal of potential hyporheic flow paths, we quantified total nitrate imported and exported for each flow path across the range of path lengths (5 to 70 m at 0.5 m increments) over the observed hydrologic conditions in 2018 (with the beaver dam) and the hypothetical hydrologic conditions in 2018 (without the beaver dam). Although the dam was present for only 68 of the 207 days within this period, its presence increases the total amount of nitrate imported into flow paths of all possible lengths, with the greatest increase occurring in short flow paths (Fig. 4). In contrast, the total amount of nitrate exported during this period depends on the flow path length. The beaver dam increases the range of net-nitrifying flow paths from 12.1 to 13.9 m, while the range of partial denitrifying paths (i.e., paths along which nitrate is not completely removed) increases from 17.6 to 25.1 m, reflecting that the dam converts flow paths in the latter range to partial exporters of nitrate (Fig. 4). Although denitrification occurs within these flow paths with the beaver dam, the shift in $Da_{DO}$ toward transport dominance results in delayed denitrification and thus only

partial removal of porewater nitrate. Flow paths between 17.6 and 25.1 m comprise approximately 15.8% of all flow paths in the floodplain (Supplementary Fig. 8), and their conversion to partial exporters of nitrate increases the range of partial nitrate exporters by 42.6% (Fig. 4). Thus, the presence of the dam, albeit transient, increases the total amount of nitrate exported over the observation period from flow paths shorter than 25.1 m, which are either net-nitrifying (for flow paths up to 13.9 m in length), partial nitrate exporters with and without the dam (paths between 13.9 and 17.6 m), or converted from completely denitrifying to partial nitrate exporters (flow paths between 17.6 and 25.1 m) (Fig. 4). Finally, the dam increases total nitrate removal along flow paths longer than 25.1 m, which completely remove all imported nitrate regardless of whether the dam is present (Fig. 4). As the majority (66.8%) of flow paths in the watershed exceed 25.1 m and are therefore net-denitrifying (remove all porewater nitrate), an increase in the density of beaver dams is likely to drive a net increase in hyporheic nitrate removal at the watershed scale.

Finally, we find that the impacts of path length—and therefore hydraulic gradient—on the hyporheic nitrate mass balance greatly exceed the impacts of variability in the rates of microbially mediated reactions. Given potential ranges in the rates of denitrification and oxygen consumption, we assessed the sensitivity of the nitrate mass balance to (1) the denitrification rate, (2) the overall rate of DO consumption, and (3) the flow path length. We employed a Morris sensitivity analysis[33,34] across the range of potential denitrification and oxygen consumption rates and three ranges of flow path lengths: 5 to 26 m; 26 to 48 m; and 49 to 70 m. Of the three parameters, path length most strongly affects the nitrate mass balance, indicating that the effects of hydraulic gradient overshadow the impacts of potential variability in reaction rates in determining the hyporheic nitrate balance (Supplementary Fig. 10). Thus, even allowing for variability and uncertainty in microbial reaction rates, beaver dams will increase nitrate removal across the watershed.

Due to the extreme hydrologic conditions that beaver dams impose, it is probable that beaver dams will overshadow future climate extremes in controlling exports of reactive N from mountain riparian zones. Growing beaver populations are likely to lead to greater hyporheic nitrate removal and reduced nitrate loading to downstream watersheds, potentially protecting freshwater quality. Our findings indicate that the impacts of beaver dams dwarf the direct hydrologic impacts of warming temperature and increased aridification, which decrease snowpack and peak discharge, on riparian water quality in

mountain watersheds. As future impacts of climate change on river hydrology and water quality are assessed, feedbacks from ecosystem changes, including those induced by management, need to be included.

# Methods

## Site description

Our field site is located within the East River watershed, near Crested Butte, Colorado, USA. The East River watershed is a mountainous, high-elevation system (2700–4100 mASL), in which hydrologic conditions are largely determined by seasonal snowmelt dynamics. Quaternary glacial soils occur throughout the watershed, underlain by Paleozic and Mesozoic sedimentary rocks, including Cretaceous Mancos Shale, with intrusions of Tertiary laccoliths. The East River floodplains consist of 1–2 m of soil above 1–4 m of alluvium, which is underlain by bedrock. Water in the subsurface is hydrologically connected to the East River.

## Collection of field samples and measurements

Field measurements and samples were collected from May through October in 2018 and 2019 at a single meander on the East River (Meander Z; Supplementary Fig. 2). Water table and river surface elevations were tracked with pressure transducers (HOBO Water Level Data Logger U20-001-01, Onset Computer Corporation). Three transects, each with three transducers, were installed within the floodplain to track subsurface hydrologic conditions (Supplementary Fig. 2). The transects were aligned with the average direction of subsurface flow. A single transducer was installed at the western edge of the site to determine lateral flow contributions from the hillside into the floodplain (Supplementary Fig. 2). The transducers were suspended on galvanized steel cables to depths of 1.5–1.7 m below ground surface within 2.0 m stilling wells (5.1 cm diameter; polyvinyl chloride (PVC); 1 mm screen slot). Two transducers were installed in the East River channel at two locations along the meander (Supplementary Fig. 2). The river transducers were suspended on galvanized steel cable in PVC well screens (1 mm slot), which were secured to fence posts driven 0.5 m into the riverbed. All transducers recorded hourly absolute pressure measurements. Absolute pressure was converted to hydrostatic pressure using barometric pressure measurements, which were recorded hourly by a transducer installed 1 m above ground surface at the site. A common datum was established at the site, and well casing elevations were measured relative to the datum using a survey-grade automatic level instrument. To convert recorded pressure to depth-to-water measurements, the distance from the top-of-casing to the water level was recorded at each transducer well one week after installation. This distance was measured twice annually to ensure transducer accuracy. Depth-to-water measurements were converted to water surface elevations.

Porewater and river water samples were collected from piezometers and the river channel using a peristaltic pump. A 2 L purge volume was collected and discarded from the piezometers prior to sampling. A multi-parameter probe was used to measure pH, electrical conductivity (EC, mS/m), and oxidation-reduction potential (ORP, mV) in the field. Dissolved oxygen (DO) was measured colorimetrically in the field on unfiltered water using the indigo carmine method (CHEMetrics, #K-7513, 1–15 ppm) or the Rhodazine D™ method (CHEMetrics, #K-7553, 0–1 ppm) and a portable spectrophotometer (CHEMetrics, Inc.). Filtered samples (0.45 mm PVDF syringe filter) were collected for quantification of anions (Cl-, $SO_4^{2-}$, $NO_3^-$), dissolved inorganic carbon (DIC), dissolved organic carbon (DOC), ammonium ($NH_4^+$), and metals. The filtered samples were shipped to the lab and stored at 4 °C prior to analyses. The 2018 porewater and river water data are published and publicly available[35], and the 2019 data will be published and available in the Watershed Function SFA, ESS-DIVE data repository.

## Laboratory analyses of field samples

Elemental concentrations were measured by inductively coupled plasma mass spectrometry (ICP-MS; Perkins-Elmer Elan DRC II) after acidification and dilution with ultrapure 0.16 M nitric acid and addition of an internal standard. Anions (Cl-, $SO_4^{2-}$, $NO_3^-$) were measured by ion chromatography (IC; Dionex ICS 2100- IC, Thermo Scientific). Ammonium samples were acidified to pH 2 with 2 M HCl and analyzed by flow injection analysis using the colorimetric salicylate method (Lachat Instruments). Total dissolved carbon and organic carbon were measured on a Shimadzu TOC-V analyzer with a nondispersive infrared detector, and dissolved inorganic carbon was determined by the difference. Total dissolved carbon was measured by catalytically aided combustion at 680 °C, and DOC was measured as nonpurgeable organic carbon, for which samples were acidified with HCl and purged with $N_2$ (g) to remove inorganic carbon prior to analysis.

## Reactive transport model development

A 1D reactive transport model was developed to represent the hydrologic and biogeochemical processes occurring along the hyporheic flow paths at the site. The model was developed using PFLOTRAN, an open-source reactive transport simulator code[36]. For the base simulations, the model domain was based on the MZA transect, which is 58 m in length, and was composed of a structured grid consisting of 580 cells, corresponding to a uniform discretization of 0.1 m along the Y axis and 1 m in both the X and Z axes. In the variable flow length simulations, the Y axis length was varied between 5 and 70 m while the discretization was unchanged from the base simulations, resulting in grids ranging from 50 to 700 cells. The maximum timestep used in all simulations was 1 h.

PFLOTRAN was run in Richards mode, which solves the Richards Equation for variably saturated flow. All simulated flow paths were fully saturated. Average hydraulic conductivity of the floodplain alluvium was measured with a permeameter, for which five cores were collected with a slide hammer and translucent polycarbonate core sleeves from a trench wall perpendicular to the direction of flow. Prior to measurement, the cores were visually inspected to confirm uniform packing. They were then loaded directly into the permeameter. The average measured hydraulic conductivity values were converted to intrinsic permeability using the dynamic viscosity and specific weight of water. A permeability of $2.26 \times 10^{-11}$ m$^2$ was used in the simulations. A porosity of 0.2 was used in the simulations and was based on the value determined by ref. 2 on similar soils from the same floodplain using pedotransfer functions.

We implemented a biogeochemical reaction network to simulate microbial and geochemical processes within the riparian zone. The reaction network consisted of aerobic microbial respiration; ammonification; nitrification; denitrification; calcite precipitation and dissolution; and acetogenesis. Dissolved organic carbon within the model was produced entirely through acetogenesis, which was simulated as the dissolution of a solid organic matter (SOM) phase to acetate, as in ref. 2. All DOC was represented as acetate. The rates of aerobic microbial respiration, denitrification, and nitrification were calculated using single Michaelis-Menten kinetic reactions applying the following general rate law:

$$R_S = \mu_{max} \cdot X_{im} \cdot \frac{C_S}{K_S + C_S} \cdot \frac{C_{TEA}}{K_{TEA} + C_{TEA}} \cdot \frac{K_I}{K_I + C_I} \qquad (1)$$

where $\mu_{max}$ is the maximum reaction rate; $X_{im}$ is the concentration of microbial biomass; $C_S$ is the substrate concentration; $C_{TEA}$ is the terminal electron acceptor concentration; $C_I$ is the concentration of the inhibitor species; and $K_S$, $K_{TEA}$, and $K_I$ are the half-saturation constants for the substrate, terminal electron acceptor, and inhibition species. Reaction stoichiometries and model parameters are shown in Supplementary Tables 1 and 2. Microbial biomass was fixed at $1 \times 10^{-5}$

mol-biomass / m³-bulk in all simulations. Biomass yield was set to zero, and no decay term was included. The maximum reaction rates for aerobic respiration, $\mu_{aer}$, and nitrification, $\mu_{nit}$, were determined by fitting the model DO output to the observations while maintaining a $\mu_{aer} : \mu_{nit}$ ratio equal to the ratio of $\Delta G_{aer} : \Delta G_{nit}$ (Supplementary Table 2). This approach ensured that partitioning of $O_{2(aq)}$ between the competing processes of aerobic respiration and nitrification was consistent with the relative energetics of these reactions[20], with aerobic respiration preferentially consuming $O_{2(aq)}$. Calcite precipitation and dissolution were modeled as kinetic processes using transition state theory (TST) rate laws with the following form:

$$R_m = -A_m \cdot \left( k_n + (k_{H^+} \cdot a_{H^+}) + \left( k_{HCO_3^-} \cdot a_{HCO_3^-} \right) \right) \cdot \left( 1 - \frac{Q}{k_{eq}} \right) \quad (2)$$

where $k_n$, $k_{H^+}$, and $k_{HCO_3^-}$ represent rate constants for neutral, acidic, and additional ($j$th) reaction mechanisms at 25 °C, respectively; $a_{H^+}$ represents proton activity and $a_{HCO_3^-}$ represents bicarbonate activity; $Q$ represents the ion activity product of the mineral phase; $A_m$ represent the calcite surface area; and $k_{eq}$ represents the calcite equilibrium constant. Dissolution of SOM was represented as an equilibrium process. Finally, ammonification was represented as the kinetically controlled dissolution of N-bearing SOM phase (N-SOM dissolution). Solid phase reaction stoichiometry and kinetic parameters are summarized in Supplementary Table 3.

Hydrostatic boundary conditions were imposed at the upstream and downstream boundaries of the model domain using observed upstream and downstream river elevations (Fig. 2). For simulations without the dam, a set of artificial river surface elevations was created by maintaining the difference between upstream and downstream water levels immediately prior to commencement of dam construction (Supplementary Fig. 11). Changes in upstream water levels over the period for which the dam was removed exactly mirror downstream water level changes, resulting in a steady gradient across the meander.

The average river water composition (Supplementary Table 4) was used as the geochemical boundary conditions at the upstream and downstream boundaries. Although the river water composition varied throughout the sampling period, we treated the composition as fixed because our modeling goal was not to exactly replicate the biogeochemical reactions occurring along the transect, but to develop a model that was representative of these processes more generally. For that purpose, using the average river water composition was appropriate.

The reactive transport model was validated against hydrologic and porewater observations from 2018 and 2019. The hydrologic component of the model produced output that was spatially and temporally consistent with observations across both years (Supplementary Fig. 12). Likewise, the reaction network yielded geochemical output that was spatially and temporally consistent with porewater observations (Supplementary Figs. 13–15).

Three sets of simulations were run: (1) base 2018 and 2019 simulations, in which the observed hydrologic conditions from 2018 and 2019 were applied as boundary conditions; (2) a 2018 no-dam simulation, in which the upstream hydrologic effects of the beaver were replaced with estimated 2018 upstream water elevations had the dam not been built; and (3) variable flow length simulations, in which the flow path length was varied between 5 to 70 m and the 2018 hydrologic conditions with and without the beaver dam were applied.

**Damköhler calculations**

Previous studies have demonstrated that net reactive N source or sink behavior of hyporheic zone flow paths is primarily a function of the transport timescale of water and the reaction timescale of DO consumption[20]. This relationship reflects that elevated DO concentrations inhibit denitrification and promote nitrification while low DO concentrations promote denitrification and inhibit nitrification. Thus, the extent to which a flow path will serve as a source or sink of reactive N will be predominantly controlled by the rates of DO supply and demand[20,21]. The Damköhler number for DO, $Da_{DO}$, defined as the ratio of the water transport timescale and the DO reaction rate timescale, is therefore a strong indicator of the potential for the flow path to be either net-nitrifying or net-denitrifying. As in ref. 20, we define the Damköhler number for DO consumption as:

$$Da_{DO} = \frac{\tau}{V_{O_2}} \quad (3)$$

where $V_{O_2}$ is the timescale of overall DO consumption, $\tau$ is the water residence time, and $\tau = \frac{L}{v}$ with $L$ being the length of the flow path and $v$ the mean advective velocity.

In our reaction network, DO is consumed by aerobic respiration and nitrification. The overall rate of DO consumption (mol-$O_2$ / s) can therefore be represented as the sum of the rates of these processes:

$$R_{O_2} = \left( \mu_{aer} \cdot X_{aer} \cdot \frac{[DOC]}{K_{DOC} + [DOC]} \cdot \frac{[O_{2(aq)}]}{K_{O_2,aer} + [O_{2(aq)}]} \right) + \left( \mu_{nit} \cdot X_{nit} \cdot \frac{[NH_{3(aq)}]}{K_{NH_{3(aq)}} + [NH_{3(aq)}]} \cdot \frac{[O_{2(aq)}]}{K_{O_2,nit} + [O_{2(aq)}]} \right) \quad (4)$$

Along a given flow path of length $L$, the overall rate of DO consumption can be calculated at any point $i$ along the flow path given the concentrations of DO, DOC, and NH₃(aq) at $i$. From the PFLOTRAN model output, we can obtain dissolved oxygen concentration, $[O_{2(aq)}]$, dissolved organic carbon concentration, $[DOC]$, and ammonia concentration, $[NH_{3(aq)}]$, at any point $i$ along the flow path, and we thus can calculate the overall rate of oxygen consumption at $i$. The timescale of DO consumption at $i$ is therefore determined as:

$$V_{O_2,i} = \frac{R_{O_2,i}}{m_{O_2,i}} \quad (5)$$

where $m_{O_2,i}$ is moles of $O_2$ at $i$. Given our model discretization of $1 \times 1 \times 0.1$ m and a porosity of 0.2, $m_{O_2,i}$ can be calculated as:

$$m_{O_2,i} = 0.1*1*1*0.2*1000*[O_{2(aq)}] \quad (6)$$

To calculate $Da_{DO}$, the transport timescale, $\tau$, at $i$ is needed, which can be calculated from the advective velocity, $v$, at $i$, which we obtain from the model output. If $i$ is the distance from the upstream boundary, then

$$\tau = \frac{i}{v}. \quad (7)$$

The Damköhler number for oxygen can therefore be calculated at any point $i$ along a flow path as:

$$Da_{DO,i} = \left( \frac{i}{v} \right) \Big/ \left( \frac{R_{O_2,i}}{m_{O_2,i}} \right) \quad (8)$$

Using this formulation, we calculated $Da_{DO}$ at each cell along the modeled flow paths. To assess whether the flow path was nitrifying or denitrifying, the $Da_{DO}$ was evaluated at the last grid cell along the flow path.

**Monte Carlo analysis of hyporheic redox zonation**

The extent of the aerobic zone is determined by the overall rate of oxygen consumption and the rate of transport (advection) of dissolved oxygen. Thus, uncertainty in these parameters, and particularly the

rate of overall oxygen consumption, will affect uncertainty in the model output and predicted redox zonation. To assess uncertainty in redox zonation arising from uncertainty in the rate of overall oxygen consumption, we performed a set of Monte Carlo simulations (5000) in which we varied the rate constants, $\mu_{max}$, for aerobic respiration and nitrification—the two oxygen-consuming reactions in our reaction network. We determined possible ranges for these rate constants by examining published studies in which the rates were either measured or simulated. For aerobic respiration, this range was $1.30 \times 10^{-4}$ to $2.01 \times 10^{-3}$ (mol m$^3_{bulk}$) / (L mol$_{bio}$ s), while for nitrification, the range was $4.68 \times 10^{-5}$ to $7.26 \times 10^{-4}$ (mol m$^3_{bulk}$)/(L mol$_{bio}$ s). We intentionally considered a broad range of possible rate constants in order to examine the relative impacts of beaver dams and reactions rates on hyporheic redox zonation. As with the base simulation, within each Monte Carlo realization, the ratio of the aerobic respiration and nitrification rates was equal to the ratio $\Delta G_{aer} : \Delta G_{nit}$ to ensure partitioning of $O_{2(aq)}$ between the competing reactions was consistent with the relative energetics of the reactions. All other model parameters were unchanged from the base simulation. Each realization was first spun up to steady state conditions over a period of 7000 h and then run over the 2018 hydrologic boundary conditions, as in the base simulation. For each model realization, we determined the distance to the point on the flow path where $Da_{DO}$ was equal to 1, which is the location where transport and reaction processes equally influence DO concentrations, and which we define as the transition between aerobic and anaerobic zones. We then plotted the distributions of these distances at the three specified time points (Supplementary Fig. 5).

**Determination of beaver dam prevalence**
Beaver dams were identified visually in a Google™ Earth satellite image of the East River watershed taken in October 2019. Only clearly identifiable beaver dams were counted. We visited nine of the 18 locations identified in the imagery to confirm the presence of a beaver dam. The satellite images showing the locations of the dams are presented in Supplementary Fig. 6. Each individual dam is shown in Supplementary Fig. 7.

**Determination of flow path length distribution**
Flow path lengths were determined using the distance measuring tool in Google Earth™. A flow path was defined as the shortest point across the riparian zone between upstream and nearest downstream riverbanks and were roughly aligned with the average valley grade. This method for determination of flow paths was based on and supported by empirical evidence from the field. Only flow paths between 5 and 70 m were considered. A histogram of measured flow path lengths is shown in Supplementary Fig. 8, and a map showing the measured flow paths is shown in Supplementary Fig. 9.

**Morris sensitivity analysis of hyporheic nitrate mass balance**
We employed a Morris sensitivity analysis to assess the sensitivity of the hyporheic nitrate mass balance to (1) the rate constant for denitrification, $\mu_{max-DEN}$, (2) the rate of overall oxygen consumption, $R_{DO}$, and (3) flow path length, $l$. The analyses were performed using the SALib software package[37,38]. The range of $R_{DO}$ over which we examined the nitrate mass balance was the same range used in the Monte Carlo analysis described above, while the range of $\mu_{max-DEN}$ ($3.13 \times 10^{-5}$ to $2.57 \times 10^{-3}$ (mol m$^3_{bulk}$)/(L mol$_{bio}$ s)) was determined from a literature analysis. We performed the sensitivity analysis over three ranges of flow path lengths: 5 to 26 m, 27 to 48 m, and 49 to 70 m. Results of the analyses are shown in Supplementary Fig. 10.

**Data availability**
The authors declare that the data supporting the findings of this study are available within the article and its Supplementary Information file.

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

## Acknowledgements

This work was supported by the Watershed Function Science Focus Area funded by the US Department of Energy (DOE), Office of Science, Office of Biological and Environmental Research (BER), under Award Number DE-AC02-05CH11231 (PN, NB); by the BER Environmental Systems Science Program under Project Award Number DE-SC0016544 (SF); and by the DOE Office of Science Graduate Student Research Program (CD). The Rocky Mountain Biological Laboratory in Gothic, CO, USA, provided field support for this research.

## Author contributions

C.D., P.N., and S.F. conceived of and designed the study. C.D. developed and managed the field site, conducted sample collected, and performed in-field measurements. Field work was aided by P.F., P.N., and S.F. Laboratory analyses were performed by P.F. C.D. developed and implemented the PFLOTRAN model, with input from N.B. D.D. helped initiate PFLOTRAN and provided hydrologic expertise on East River. The manuscript was written by C.D. with N.B., P.N., and S.F.

## Competing interests

The authors declare no competing interests.
