## [Peer Review File · Nature Communications]

Beaver dams overshadow climate extremes in controlling riparian hydrology and water qualityEditorial Note: In their review of the first version of this manuscript, Reviewer #2 added their comments to an annotated manuscript file. These comments, excluding minor textual revisions, have been addressed by the authors in their first rebuttal and are included in this Peer Review File.

REVIEWER COMMENTS

Reviewer #1 (Remarks to the Author):

The manuscript “Beaver dams overshadow climate extremes in controlling riparian hydrology and water quality” investigated the impacts of beaver dams on riparian hydrological and nitrogen cycling at both plot and catchment scale, which is overall an interesting contribution to the hydrological and water quality communities. The expansion of the riparian aerobic zone due to beaver dam is particularly of my interests and potentially to other water quality modelers, as the redox changes due to elevations change can be significant as is proved in this study, but has rarely been incorporated into modelling in practice. However, several aspects need to be further refined.

The first aspect is the model uncertainty. In this study, the authors used a 1D reactive transport model to simulate the water and nitrogen fluxes within riparian flow paths. However, when modelling, it is always recommended to clarify the uncertainty, i.e., to what extent we can trust the results. I noticed the several assumptions made for modelling have been justified in method section; but still it is better to further clarified the modelling uncertainty even with such assumptions.

Another methodology-related issue is about the flow paths at catchment scales. It is unclear for me in terms of how they were determined. Moreover, I wonder when determining the flow paths, has DEM been taken into consideration? As far as I know, the flow paths were determined according to surface elevation and slope in most distributed hydrological models. From Figure S8, I can roughly say that DEM were not considered as the flow paths were all in straight lines. Can you also clarify this, or just state the uncertainty of not considering DEM?

The last comment is a bit personally picky. Though the language is good, the storyline of the manuscript can be further improved. E.g., when extending the analysis from case study into catchment scale, I would expect to see the necessity of such upscaling. Adding these sentences help connect different paragraphs/sections logically and build a clear storyline.

Detailed comments:

“The beaver dam increased the hydraulic gradient across the riparian zone by 161.5%, which more than doubled the extent of dissolved oxygen intrusion”

Delete “which” or add “are” behind which.

“By shortening water residence time, the beaver dam increases DO advection, promoting aerobic nitrification and mineralization of N-bearing soil organic matter^{27,28}, while suppressing the use of nitrate as a terminal electron acceptor³³.”

Water residence time of where? Please clarify.

“Where DO concentrations are controlled by advection, nitrate concentrations increase, while ammonium and DO concentrations decrease.”

This sentence is unclear to me. I can understand nitrate and ammonium respectively increase and decrease under less reduced conditions. But DO concentrations should also be higher in this context when they are controlled by advection? Moreover, the section title should contain oxygen level and nitrate transformation, since they are both introduced here.

“Nitrate removal reflects the hydraulic gradient, except that it lags by 4-5 weeks (Figure 3; Figure 1), as nitrate must first travel across the aerobic zone, where denitrification is suppressed and aerobic nitrification occurs, before it is reduced and removed from porewater.”

Does it mean the residence time of lateral inflow penetrating through the aerobic zone is 4-5 weeks?

We determined an approximate distribution of flow path lengths within a representative portion of the East River floodplain (Figures S7 and S8)

It is unclear to me in terms of how these flow paths are determined?

Figure S6. The resolution could be improved.

Figure S8 gives the full picture of flow paths at the catchment-scale. But it is not cited in the main text.

In method section "The transects were aligned with the average direction of subsurface flow"

How to know the direction of subsurface flow? By ground elevation?

"and the 2019 data will published shortly"

Change to will be published.

"Previous studies have demonstrated that net reactive N source or sink behavior of hyporheic zone flow paths is primarily a function of the transport timescale of water and the reaction timescale of DO consumption"

A citation is needed here.

Reviewer #2 (Remarks to the Author):

The manuscript "Beaver dams overshadow climate extremes in controlling riparian hydrology and water quality" explores and presents key findings of the ecological and environmental impacts introduced by beaver colonization on nutrient retention and riparian exports. The authors leveraged both measured and model datasets to provide insights into the effect of beaver dams in shifting the aeration zones and establishing critical riparian flow path lengths necessary to control reactive nitrate. The manuscript is well written and presents findings of significant interest to the scientific community. I believe that the manuscript would benefit from further clarification, particularly in the terminology, description of site conditions, and justification of assumptions used for the larger analysis. Some more detail statements about the modelling, particularly uncertainties would also help. I would recommend major revisions to improve the manuscript. Detailed comments can be found in the attached PDF.

Beaver dams overshadow climate extremes in controlling riparian hydrology and water quality

Response to Reviewers

Reviewer 1

Reviewer comment 1: (lines 20-22) Delete 'which' or add 'are' behind 'which'

Response: We agree that our syntax may have been unclear here.

Changes made: This sentence was removed in the process of re-writing the abstract.

Reviewer comment 2: (lines 94-96) Water residence time of where? Please clarify.

Response: We agree this sentence needed clarification.

Changes made: This sentence was removed in the process of revising the manuscript. Now, the initial sentences of this paragraph read: “The expansion of the aerobic zone is paired with a narrowing of the denitrification zone. Within the expanded aerobic zone, nitrate concentrations increase due to mineralization of N-bearing soil organic matter (N-SOM) and nitrification.”

Reviewer comment 3: (lines 97-99) This sentence is unclear to me. I can understand nitrate and ammonium respectively increase and decrease under less reduced conditions. But DO concentrations should also be higher in this context when they are controlled by advection? Moreover, this section title should contain oxygen level and nitrate transformation, since they are both introduced here.

Response: We agree that this sentence needs refinement. Within the aerobic zone, advection more strongly influences dissolved oxygen concentrations than reaction processes (primarily aerobic respiration), as our Damköhler analysis shows. However, reactions which consume dissolved oxygen definitively occur within the aerobic zone, decreasing its concentration. The Damköhler analysis reflects that transport processes predominate but that reaction processes nonetheless contribute to DO concentrations.

Changes made: We changed ‘control’ to ‘predominantly influence’. We also changed the section title to ‘Redox zonation during hydrologic extremes.’

Reviewer comment 4: (lines 109-111) Does it mean the residence time of lateral inflow penetrating through aerobic zone is 4-5 weeks?

Response: The lag between nitrate removal and the hydraulic gradient reflects a history of dynamic nitrate and oxygen advection, nitrate and oxygen consumption (via denitrification and aerobic metabolism, respectively), and nitrate production (via nitrification). Because we impose transient hydrologic boundary conditions in our model (corresponding to river surface elevation upstream and downstream of our monitoring transect), advection changes throughout our simulation, reflecting changing hydraulic gradients. These changes in advection alter the rates of

the reactions involving nitrate and oxygen. Thus, the lag we note is not strictly the residence time of nitrate. Our Damköhler analysis quantifies the relative contributions of transport and reaction processes in determining the concentrations of oxygen and nitrate across changes in hydraulic gradient and its impact on advection.

Changes made: No changes made.

Reviewer comment 5: (lines 132-136) It is unclear to me in terms of how these flow paths are determined?

Response: Flow path lengths were determined by measuring the distance between the upstream and downstream banks of intra-meander regions of the floodplain using satellite imagery. All flow paths were assumed to be roughly parallel to the overall valley grade. This assumption was based on extensive empirical evidence from the site. Flow paths used to derive the distribution are shown in in Figure S9. Each line in Figure S9 represents a flow path.

Changes made: In the Results and Discussion section, we clarified that satellite imagery was used to determine the flow path distribution in the paragraph beginning “The impacts of hydrologic extremes on reactive nitrogen fluxes...”. We also clarified that we assume that the flow paths are roughly parallel to the average valley grade.

Reviewer comment 6: (Figure S6) The resolution could be improved.

Response: It would be ideal to include high-resolution images of the beaver dams. However, we are limited to the inherent resolution of the satellite imagery. Although this resolution is not ideal, it is nonetheless sufficient for identifying beaver dams within the floodplain.

Changes made: No changes made.

Reviewer comment 7: (Figure S8) Figure S8 gives the full picture of flow paths at catchment-scale. But it is not cited in the main text.

Response: We refer to Figure S8 on line 134 in the original version of the manuscript.

Changes made: No changes made.

Reviewer comment 8: (line 197) How to know the direction of subsurface flow? By ground elevation?

Response: The direction of subsurface flow is assumed to be parallel to the average valley grade and is substantiated by empirical observation at the field site. However, we note that the comparison between beaver dam and seasonal hydrologic conditions across flow path lengths does not require exact determination of the direction of subsurface flow. Rather, the comparison requires a reasonable distribution of flow path lengths, which we obtained using satellite imagery (as described in our response to comment 5). The comparison between beaver dam and seasonal hydrologic conditions will be valid regardless of whether a flow path is exactly aligned with the average valley grade.

Changes made: We have clarified our use of satellite imagery and means to assessing flow paths.

Reviewer comment 9: (line 218) Change to will be published

Response: We agree with the reviewer's suggestion to change the wording.

Change made: We removed 'shortly' from the referenced sentence.

Reviewer comment 10: (lines 300-302) A citation is needed here.

Response: We agree a citation is needed.

Changes made: We added the following citation:

Zarnetske, et al. (2012), Coupled transport and reaction kinetics control the nitrate source-sink function of hyporheic zones, *Water Resour. Res.*, 48, W11508.

Reviewer 2

Reviewer comment 1: (line 37) jeopardizing what? ecosystem growth/development? clarify

Response: We agree that this sentence should be clarified.

Changes made: We changed the sentence so that it reads: ‘... riparian exports of nutrients and contaminants will shift in tandem, potentially jeopardizing downstream water quality.’

Reviewer comment 2: (line 40) largely

Response: We corrected this typo.

Changes made: We changed ‘largey’ to ‘largely.’

Reviewer comment 3: (line 49) awkward wording

Response: This sentence was removed in revising the manuscript.

Changes made: We deleted this sentence.

Reviewer comment 4: (line 50) Coupled effects?

Response: Because we changed ‘climate extremes’ to ‘seasonal hydrologic extremes’ in this sentence (see response to comment 5, below), adding ‘coupled’ is no longer necessary.

Changes made: The sentence was revised.

Reviewer comment 5: (line 51) This needs further clarification that leverages historical extremes, rather than projection of future climate extremes

Response: The record of peak discharge at the Almont gauge, which spans 86 years, indicates that hydrologic conditions in 2018 and 2019 are low- and high-end members, respectively. Peak discharge in 2019 exceeded 85% of recorded peak discharges, while peak discharge in 2018 was lower than 95% of recorded peak discharges. Multiple studies predict that future climate conditions will result in decreased stream flow in the Colorado Rockies (e.g., Foster et al. *Environ. Res. Lett.* **10** (2016); Christensen et al. *Ecohydrology* **14** (2021); Siirila-Woodburn et al. *Nat. Rev. Earth Environ.* **2** (2021)). Thus, we suggest that 2018 is indeed a reasonable analog for future conditions. However, we acknowledge that we compared historical seasonal extremes to beaver dam conditions.

Changes made: We changed ‘climate extremes’ to ‘climatic hydrologic extremes’ in the sentence beginning ‘In this study, we compare the effects of beaver dams to ...’ and we reference the assessment to past extremes while noting their potential representation of future changes. Importantly, we note that the beaver influences on hydraulic extremes are so vast that climate changes will always be dwarfed by comparison.

Reviewer comment 6: (line 51) I don't think it is accurate to classify a low peak flow and a high peak flow year as "climate extremes" from the information provided. Further information such as the classification of annual precipitation index would help to further justify that these juxtaposed years represent extreme differences.

Response: As described in our response to comment 5, the 86-year discharge record at the Almont gauge clearly shows that discharge in 2018 and 2019 are low- and high-water endmembers. Given that multiple studies predict decreased discharge under future climate conditions, the low water conditions in 2018 are a reasonable analog for future climate extremes. Moreover, and as noted above, the beaver dams lead to extremes that vastly outweigh climatic variation—past or future.

Changes made: We added the following sentence to clarify that hydrologic conditions in the East River watershed mirror regional trends in 2018 and 2019: 'Hydrologic conditions throughout the East River watershed reflected the regional trends in these years (Figure S1).'

Reviewer comment 7: (line 60) across riparian areas?

Response: We agree that the wording of this sentence could be improved.

Changes made: We changed the final clause of the sentence so that it now reads: "...leading to shortened water residence times and increased oxygen and nitrogen fluxes across hyporheic zones."

Reviewer comment 8: (line 66) Clarification needs to be made throughout this section on the status of the beaver dam. Based on the rapid change in water level I suspect that the beaver dam was destroyed, but was not able to confirm this in the text. This limits some of the capability for interpretation of results as if the beaver dam remains I don't believe a direct comparison of 2018 and 2019 is valid as 2019 would be a full year of beaver dam effects

Response: The dam was breached and completely washed away on October 5, 2018. There were no remnants of the dam in 2019. We agree that it is important to clarify that the dam was not present in 2019.

Changes made: We added the following sentence describing the timeframe over which the dam was present at the site: "Construction of the dam began in late July 2018, and the dam was breached and destroyed on October 5, 2018."

Reviewer comment 9: (lines 71-81) Further explanation needed here on the transient nature of the impact. Was the beaver dam destroyed?

Response: Yes, the beaver dam was destroyed on October 5, 2018. As in our response to comment 8 (above), we agree that clarification of the dam's status is needed.

Change made: The change made to address comment 8 also addresses this comment.

Reviewer comment 10: (lines 83-92) How was the significance quantified? To what level?

Response: We agree with the reviewer that our language in this sentence regarding significance was unclear.

Changes made: We changed the sentence so that it now reads: ‘The beaver-driven hydraulic gradient more than doubles the extent of the riparian aerobic zone relative to its extent during seasonal extremes.’

Reviewer comment 11: (line 88) Can you provide uncertainty ranges on these estimates? Based on the model results, there are increased simulated gradients and likely higher advection velocities. This will shift the Da estimation.

Response: We agree that it is important to address uncertainty in our analysis of subsurface redox (aerobic v. anaerobic) zonation and its response to hydrologic extremes. The extent of the aerobic zone is determined by the overall rate of oxygen consumption (reaction rate) and the advective flow rate along the transect (transport rate). Thus, uncertainty in the predicted extent of the aerobic zone arises from uncertainty in the reaction and transport rates in the model.

Uncertainty in the reaction rate is much greater than uncertainty in the transport rate. The parameters that define the transport rate within the model are the permeability and porosity of the flow path and the hydraulic potentials at the model boundaries, all of which we measured at our site. Given the accuracy of the pressure transducers and the hourly frequency of our measurements, the measured hydraulic potentials at the transect boundaries are unlikely to introduce significant uncertainty to the model output. Similarly, our measurements of soil characteristics (hydraulic conductivity, porosity) are unlikely to introduce significant uncertainty. The fidelity of predicted hydraulic potentials to measured potentials across the transect (Figure S10) supports the conclusion that uncertainty in the transport rate is minimal.

In contrast to the transport rate, the parameters that define the rate of oxygen consumption (the rates of aerobic respiration and nitrification) cannot reasonably be measured along the transect; however, changes in dissolved oxygen concentrations were measured. The oxygen consumption reactions were deduced by fitting reaction parameters to the measure DO concentrations. Because our measured oxygen concentrations are temporally limited relative to the hydraulic potentials, the rate of oxygen consumption has greater uncertainty than transport rate.

To assess the impact of uncertainty in the rate of oxygen consumption on the extent of the aerobic zone, we performed a set of Monte Carlo simulations (5000 model realizations) in which we varied the rates of aerobic respiration and nitrification. Based on previously published studies, we determined the possible range of maximum aerobic respiration rates and sampled from this range assuming a uniform distribution. We intentionally examined a broad range of possible rate constants, as our overall goal is not to exactly replicate conditions at our study site, but to more broadly probe the impacts of beaver dams on the hyporheic processes. As described in the Methods section of the manuscript, the partitioning of DO between the competing processes of aerobic respiration and nitrification reflect the relative energetics of these reactions. Thus, the nitrification rate in the model was determined by multiplying the randomly sampled aerobic respiration rate by the ratio of the Gibbs free energy of the nitrification reaction (-181 kJ

/ mol C) to the Gibbs free energy of aerobic respiration (-501 kJ m / mol C) (i.e., by a factor of 0.361) (reaction rate formulations are defined in Table S1).

For each model realization in the Monte Carlo simulation, we compared the dissolved oxygen concentrations at 3 and 16 m along the flow path to the oxygen concentrations at MZA1-1 and MZA1-2 as measured on 5-21-2018, 7-24-2018, and 10-2-2018 (the dates correspond, respectively, to seasonal high-water conditions, extreme low-water conditions, and the beaver-driven high-water conditions). For each valid simulation, we determined the distance to the point on the flow path where the Damkohler number for dissolved oxygen was equal to 1 (where transport and reaction processes exert equal influence on dissolved oxygen concentrations). As described in the main text of the manuscript, we define the point at which the Damkohler number for dissolved oxygen equals 1 as the transition from aerobic and anaerobic conditions. We then plotted the distribution of this distance for all valid simulations at seasonal high-water conditions, extreme low-water conditions, and the beaver-driven high-water conditions (Figure S5).

We find that the relative shifts in the extent of redox zonation between the seasonal and beaver-drive hydrologic extremes is unchanged, as the shifts in the distributions of the distances to $Da_{DO} = 1$ are proportional to the shifts in the hydraulic gradient.

Changes made: We performed a Monte Carlo analysis (5000) simulations to assess the impact of the rate of overall oxygen consumption on redox zonation along the flow path. We revised the manuscript to include this analysis.

Reviewer comment 12: (line 105) Following from the previous section, it is not clear what the impacts of the beaver dam were in 2019 (i.e., if it was still there)

Response: The dam was washed away on October 5, 2018. We agree that clarification of the dam's status is needed.

Changes made: The changes made in response to comment 8 address the issue raised in this comment.

Reviewer comment 13: (lines 107-108) Add the description of steady-state estimation here.

Response: We agree that a description of the steady-state conditions is needed.

Changes made: We added the following sentence to clarify the steady-state conditions we used.

“In the steady-state simulation, the hydraulic gradient was set to the mean of the hydraulic gradients across the riparian zone on the first days in 2018 and 2019 when the river channel was free of ice (April 7 and April 21, respectively).”

Reviewer comment 14: (lines 107-108) What are 'initial' gradients? The first available measurement? Or what period? Were the periods equal (i.e., both start in May)? This is difficult to interpret with the information provided as no baseflow conditions (short or more preferably

long-term) are provided to give context. Even historical winter precipitation data would be useful for characterization.

Response: We agree that additional clarification is needed about the initial gradients and the length of the measurement periods.

The initial gradients were the gradients across the riparian zone as measured on the first full days in 2018 and 2019 when the river channel was free of ice (April 7 and April 21, respectively).

Pressure data was collected continuously over from October 15, 2017, until October 2, 2019. However, we only considered pressure data that were collected when the river channel was free of ice, and the first date on which this was true differed between years. Further, in the inter-annual comparisons presented in Figure 3, we do not use data collected after October 31, 2018, so that the datasets from the two years are of comparable length. Comparison of cumulative nitrate removal (Figure 4) were made over equivalent durations (3993 h).

Changes made: The changes made to address comment 13 also address this comment.

Reviewer comment 15: (lines 122) Google Earth

Response: We agree with the reviewer's suggestion to identify the source of the satellite imagery we used.

Changes made: We changed 'satellite imagery' to 'Google Earth imagery.'

Reviewer comment 16: (lines 132-136) PFLOTRAN for each site, or leveraging output from the study site?

Response: For each flow path within the distribution, we created a new 1D PFLOTRAN model in which the model domain was altered to reflect the length of the flow path. Other model parameters were unchanged from the simulation of the representative site. Thus, output unique to each flow path was obtained and subsequently assessed in the large-scale analysis.

Changes made: We have added language clarifying that a reactive transport model was created for each flow path within the distribution.

Reviewer comment 17: (lines 132-136) In terms of elevation difference from upstream to downstream, how representative is the study site compared to other sites? Is there an equivalence in topographic relief?

Response: The topographical gradient at the study site is representative and consistent with the average grade in the floodplains throughout the watershed. This assessment is based on a digital elevation model of the East River watershed ("LiDAR collection in August 2015 over the East River watershed." <https://doi.org/10.21952/WTR/1412542>)

Changes made: We clarified that the average gradient at the study site representative of meandering reaches throughout the watershed and that this assessment is based on a digital elevation model of the watershed.

Reviewer comment 18: (line 139) How uncertain is this given potential model uncertainty (as in previous comment) and with the assumption of equivalent hydraulic gradients (again as with previous comment of site representativeness)?

Response: The topographical relief at the study site is representative of the average relief and gradients in floodplains throughout the watershed. However, differences in relief do not affect our findings, as we are ultimately assessing the impacts of beaver dams relative to non-dam conditions – whatever the non-dam conditions may be. Further, our analysis of the hyporheic nitrate mass balance for flow paths ranging in lengths is an assessment of the impact on hydraulic gradient, as the hydraulic gradient is a function of both the head drop (i.e., rise in upstream water level due to beaver dam) and the flow path length. If gradients established without dams are higher (or lower) relative to the gradients at our site, the beaver dam-induced gradients are also higher (or lower) by a comparable amount, and the relative impacts of the dam are described by our analysis across the distribution of flow path lengths.

To assess the uncertainty of flow path length and the rates of overall oxygen consumption and denitrification on the hyporheic mass balance, we performed a Morris sensitivity analysis on the cumulative nitrate mass balance between April 7 and October 31, 2018 (timeframe include the beaver dam). The results of the sensitivity analysis are shown in Figure S10 and reveal that flow path length, and therefore hydraulic gradient, most strongly affects the hyporheic nitrate mass balance. The results support our floodplain-scale analysis, which demonstrates that the impact of beaver dams will vary depending on the length of the flow paths within the floodplain.

Changes made: We performed a Morris sensitivity analysis on the cumulative hyporheic nitrate mass balance from April 7 through October 31, 2018. We assessed the sensitivity of the nitrate mass balance to (1) the rate of overall oxygen consumption; (2) the rate of denitrification; and (3) the length of the flow path, and therefore the hydraulic gradient. We included a discussion of our analysis and the results in the main text of the manuscript. The plots of the results are shown in Figure S10.

Reviewer comment 19: (Figure 3 caption) This description should be in the main text

Response: We agree that the description of the steady-state conditions should be in the main text.

Changes made: The changes made in response to comment 13 directly address this comment.

Reviewer comment 20: (Figure S2) A scale on figure S2 would be helpful

Response: We agree.

Changes made: We added a scale bar to Figure S2, as well as a north arrow.

REVIEWER COMMENTS

Reviewer #1 (Remarks to the Author):

The authors addressed my comments well, and I have no further comments. Thank you for contributing to the field of hydrology.